# Molecular mechanism of Arp2/3 complex inhibition by Arpin

Fred E. Fregoso [1,5], Trevor van Eeuwen [1,4,5], Gleb Simanov[2], Grzegorz Rebowski [1], Malgorzata Boczkowska [1], Austin Zimmet[1], Alexis M. Gautreau[2,3] & Roberto Dominguez [1✉]

Positive feedback loops involving signaling and actin assembly factors mediate the formation and remodeling of branched actin networks in processes ranging from cell and organelle motility to mechanosensation. The Arp2/3 complex inhibitor Arpin controls the directional persistence of cell migration by interrupting a feedback loop involving Rac-WAVE-Arp2/3 complex, but Arpin's mechanism of inhibition is unknown. Here, we describe the cryo-EM structure of Arpin bound to Arp2/3 complex at 3.24-Å resolution. Unexpectedly, Arpin binds Arp2/3 complex similarly to WASP-family nucleation-promoting factors (NPFs) that activate the complex. However, whereas NPFs bind to two sites on Arp2/3 complex, on Arp2-ArpC1 and Arp3, Arpin only binds to the site on Arp3. Like NPFs, Arpin has a C-helix that binds at the barbed end of Arp3. Mutagenesis studies in vitro and in cells reveal how sequence differences within the C-helix define the molecular basis for inhibition by Arpin vs. activation by NPFs.

[1] Department of Physiology and Biochemistry and Molecular Biophysics Graduate Group, Perelman School of Medicine, University of Pennsylvania, Philadelphia, PA 19104, USA. [2] Laboratoire de Biologie Structurale de la Cellule, CNRS, Institut Polytechnique de Paris, 91128 Palaiseau, France. [3] Skolkovo Institute of Science and Technology, 121205 Moscow, Russia. [4] Present address: Laboratory of Cellular and Structural Biology, The Rockefeller University, New York, NY 10065, USA. [5] These authors contributed equally: Fred E. Fregoso and Trevor van Eeuwen. ✉email: droberto@pennmedicine.upenn.edu

Arp2/3 complex is an actin filament nucleation and branching complex of seven proteins, including two actin-related proteins, Arp2 and Arp3, and five scaffolding subunits, ArpC1 to ArpC5[1]. Branched actin networks nucleated by Arp2/3 complex are abundant near the cell periphery and around most intracellular membranous organelles, where they drive a host of cellular processes such as cell motility, vesicular trafficking, membrane scission, and mechanosensation[2,3]. Several proteins regulate these processes by activating or inhibiting Arp2/3 complex and by stabilizing or disassembling branched networks. Among these proteins, WASP-family nucleation-promoting factors (NPFs) activate Arp2/3 complex by inducing a conformational change that prompts binding of the complex to the side of a pre-existing (mother) filament and formation of a new (branch) filament that grows at a ~70° angle relative to the mother filament[4–8]. A positive feedback loop regulated by the small GTPase Rac and involving Arp2/3 complex activation by the NPF WAVE controls the assembly of branched networks at the cell cortex to drive cell motility[9]. Rac also regulates the activity of the Arp2/3 complex inhibitor Arpin[10]. Arpin inhibition of Arp2/3 complex is thus thought to act as a brake on the Rac-WAVE-Arp2/3 complex feedback loop to control the directional persistence of cell migration[9]. Consistent with this idea, Arpin depletion results in faster protruding lamellipodia and increased cell migration, whereas intracellular injection of Arpin destabilizes cell protrusions and reduces cell migration[10,11].

While recent cryo-electron microscopy and tomography (cryo-EM and cryo-ET) structures illuminate the molecular mechanisms of Arp2/3 complex activation by WASP-family NPFs and branch formation[7,8], the mechanism of inhibition by proteins such as Arpin is unknown. Here, we describe the cryo-EM structure of Arpin bound to Arp2/3 complex at 3.24-Å resolution. Unlike NPFs that bind to two sites on Arp2/3 complex, one on subunits Arp2-ArpC1 and one on Arp3, Arpin binds only to Arp3. However, the interactions of Arpin and NPFs with Arp3 are very similar, involving C-terminal Central (C) and Acidic (A) domains. Most of the C domain is folded as a helix (C-helix). The C-helix of NPFs binds at the barbed end of both Arp2 and Arp3, whereas that of Arpin has evolved to bind specifically to Arp3 and not Arp2. The analysis of Arpin mutants in cells and NPF-Arpin hybrid constructs in vitro reveals how differences in the sequence of the C-helix of Arpin limits binding to Arp3 and how this leads to inhibition. Contrary to the initially proposed model of Arpin inhibition through direct competition with NPFs, the structural-functional data presented here support a model in which Arpin competes only with NPF binding to Arp3 and locks Arp2/3 complex in the inactive conformation through interactions of the C-helix with the C-terminal inhibitory tail of Arp3.

## Results

**Arpin binds to a single site on subunit Arp3 of Arp2/3 complex.** Arpin was discovered through a bioinformatics search for proteins containing a C-terminal A domain analogous to that of NPFs (Fig. 1a), and thus when it was revealed that Arpin inhibited Arp2/3 complex it was assumed to be through competition with NPFs[10]. Because NPFs bind to two sites on Arp2/3 complex[6], Arpin was also thought to bind to two sites[12], as further suggested by a low-resolution negative stain EM study[13]. Here, we set out to determine the high-resolution cryo-EM structure of Arpin bound to Arp2/3 complex. Bovine Arp2/3 complex was mixed with full-length human Arpin and the resulting complex was isolated using glycerol gradient centrifugation (Supplementary Fig. 1a). Cryo-EM data were collected on this complex and the structure was determined at 3.6-Å resolution (Fig. 1b, Supplementary Fig. 1b, c, and Methods). A

previous low-resolution small-angle x-ray scattering (SAXS) study showed that full-length Arpin consists of an N-terminal ~21 kDa globular domain and a C-terminal extension that projects from the globular domain[12]. Surprisingly, however, the cryo-EM map showed all the subunits of Arp2/3 complex, which adopts the canonical inactive conformation[14], but no extra density that could be assigned to the globular domain of Arpin was observed in either 2D class averages or the 3D reconstruction (Supplementary Fig. 1b, c). Yet, closer inspection revealed density on Arp3 that could not be accounted for by any of the Arp2/3 complex subunits and which was thus assigned to Arpin. The Arpin density occupied the cleft at the barbed end of Arp3 and extended along the Arp3 surface to engage the pocket that binds the conserved tryptophan of the A domain of NPFs. This density was very similar to that observed for NPF-binding site-2 on Arp3 in our recent cryo-EM structure of N-WASP WCA (a fragment comprising the C-terminal WH2, C, and A domains of N-WASP) bound to Arp2/3 complex[7]. Comparison of cryo-EM maps low-pass filtered to 5-Å resolution of Arp2/3 complex alone[15] and with bound Arpin or N-WASP WCA[7] further confirmed the presence of Arpin density on Arp3 and the lack of density associated with NPF-binding site-1 on Arp2-ArpC1 (Fig. 1b–d). By analogy with NPFs, the density on Arp3 was assigned to Arpin's C-terminal residues S194-D226 and comprises both C and A domains. While the existence of the A domain in Arpin was already known[10,12], that of the C domain was unexpected.

Taken together, the results described above suggest that Arpin binds through a CA region analogous to that of NPFs to a single site on Arp3 and that the globular domain does not participate in the interaction. This was a surprising finding since NPFs bind to two sites on Arp2/3 complex, and NPF-binding site-1 on Arp2-ArpC1 has higher affinity than site-2 on Arp3[16,17]. Our findings also contradicted previous negative stain EM results suggesting two Arpin binding sites on Arp2/3 complex and the participation of Arpin's globular domain in the interaction[13]. Therefore, we used isothermal titration calorimetry (ITC) to verify the 1:1 stoichiometry of Arpin binding to Arp2/3 complex, establish the affinity of the interaction, and test the potential involvement of Arpin's globular domain in binding to Arp2/3 complex. The ITC titration of full-length Arpin into ATP-bound Arp2/3 complex produced an exothermic reaction that fit best to a one-site binding isotherm ($N = 1.06$) with $K_D \sim 1.24$ μM (Fig. 1e). Very similar results were obtained with ADP-bound Arp2/3 complex (Fig. 1f), suggesting that Arpin binds Arp2/3 complex in a nucleotide-independent manner. These experiments confirmed the 1:1 stoichiometry of Arpin binding to Arp2/3 complex observed in the structure. To further verify that NPF-binding site-1 is not implicated in Arpin binding, we pre-saturated this site with actin-GCA as previously described[7,16]. Briefly, GCA is a construct in which the WH2 domain (W) of N-WASP WCA is replaced with gelsolin segment-1 (G), which has higher affinity for actin than the WH2 domain[18,19]. A 1:1:1 complex of actin:GCA:Arp2/3 complex can be purified by gel filtration[16] or glycerol gradient centrifugation[7]. Here, we used gel filtration (see Methods). During purification, actin-GCA remains bound to the higher-affinity site-1 but falls of the lower affinity site-2[16]. Remarkably, the titration of Arpin into this half-saturated complex did not alter the stoichiometry nor the binding affinity of Arpin for Arp2/3 complex (Fig. 1g), consistent with it binding to site-2 (Arp3) and not to the higher-affinity NPF-binding site-1 (Arp2-ArpC1) occupied by actin-GCA in this experiment. We finally found that the affinity and stoichiometry of the interaction were approximately the same for full-length Arpin and the C-terminal CA fragment resolved in the cryo-EM structure (residues S194-D226), confirming that the globular domain does not participate in the interaction (Fig. 1h). The CA fragment

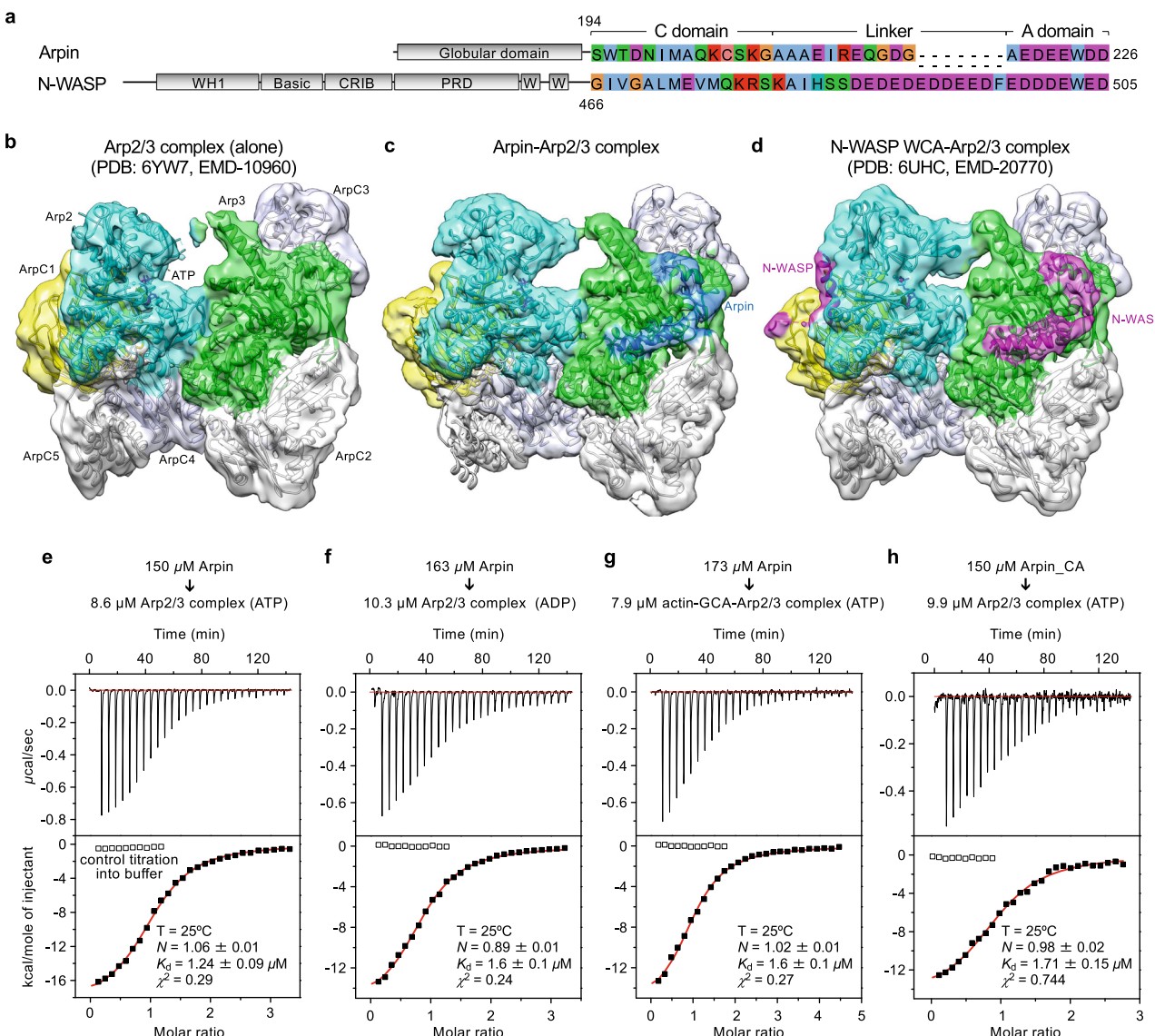

**Fig. 1 Arpin binds to a single site on subunit Arp3 of Arp2/3 complex. a** Comparison of the domains and sequences of N-WASP (NPF) and Arpin (inhibitor), both of which contain C-terminal CA regions (UniProt accession codes: human N-WASP, O00401 and human Arpin, Q7Z6K5). **b–d** Comparison of cryo-EM maps (and ribbon diagrams) low-pass filtered at the 5-Å resolution of Arp2/3 complex alone[15] (**b**) and with bound Arpin (**c**) and N-WASP CA[7] (**d**). Arp2/3 complex subunits are labeled in part b, and colored as follows: Arp2, cyan; Arp3, green; ArpC1, yellow; ArpC2, gray; ArpC3, blue-white; ArpC4, blue-white; ArpC5, gray. Arpin and N-WASP are colored marine blue and magenta, respectively. Note that the Arpin_CA and N-WASP CA densities on Arp3 are very similar. Panels **b–d** were generated with the program Chimera[41], and the maps were contoured at 4σ. **e–h** ITC titrations of full-length Arpin or Arpin_CA (in the syringe) into Arp2/3 complex (in the cell) in the ATP or ADP-bound states (as indicated). For each titration, the experimental conditions, including temperature (T) and protein concentrations, and the fitting parameters, including stoichiometry (N), dissociation constant ($K_d$), and goodness of the fit ($\chi^2$) are shown.

(construct Arpin_CA) is therefore used in most of the experiments described below.

**Cryo-EM structure of Arpin_CA bound to Arp2/3 complex shows differences with NPFs.** The structure of full-length Arpin bound to Arp2/3 complex did not provide important details, such as the exact location of Arpin's side chains, to understand its specificity for Arp3 and how this leads to inhibition. Because the globular domain of Arpin is not necessary for binding to Arp2/3 complex and was disordered in the structure, we thought that a more stable complex for cryo-EM analysis could be obtained using Arpin_CA. The structure of this complex was determined at 3.24-Å resolution (Supplementary Fig. 2), and as anticipated the map was much better defined overall than that of full-length

Arpin, allowing for the assignment of Arpin's side chains (Fig. 2 and Table 1). The Arpin_CA density on Arp3 was split into two well-defined segments, one comprising the helix of the C domain (C-helix) and the other centered around the conserved trypto-phan (W224) of the A domain (Fig. 2c). While the flexible linker between these two regions was not well-resolved in the high-resolution map, the entire CA polypeptide was clearly visualized in the map low-pass filtered to 5.5-Å resolution (Supplementary Fig. 3a). As observed with full-length Arpin, Arpin_CA only bound Arp3, which is also consistent with the ITC results (Fig. 1h) and confirms that the specificity of the interaction is fully contained within this region. This raises two questions: why does Arpin bind Arp3 and not Arp2-ArpC1 like N-WASP? and how does Arpin binding to Arp3 explain its inhibitory activity?

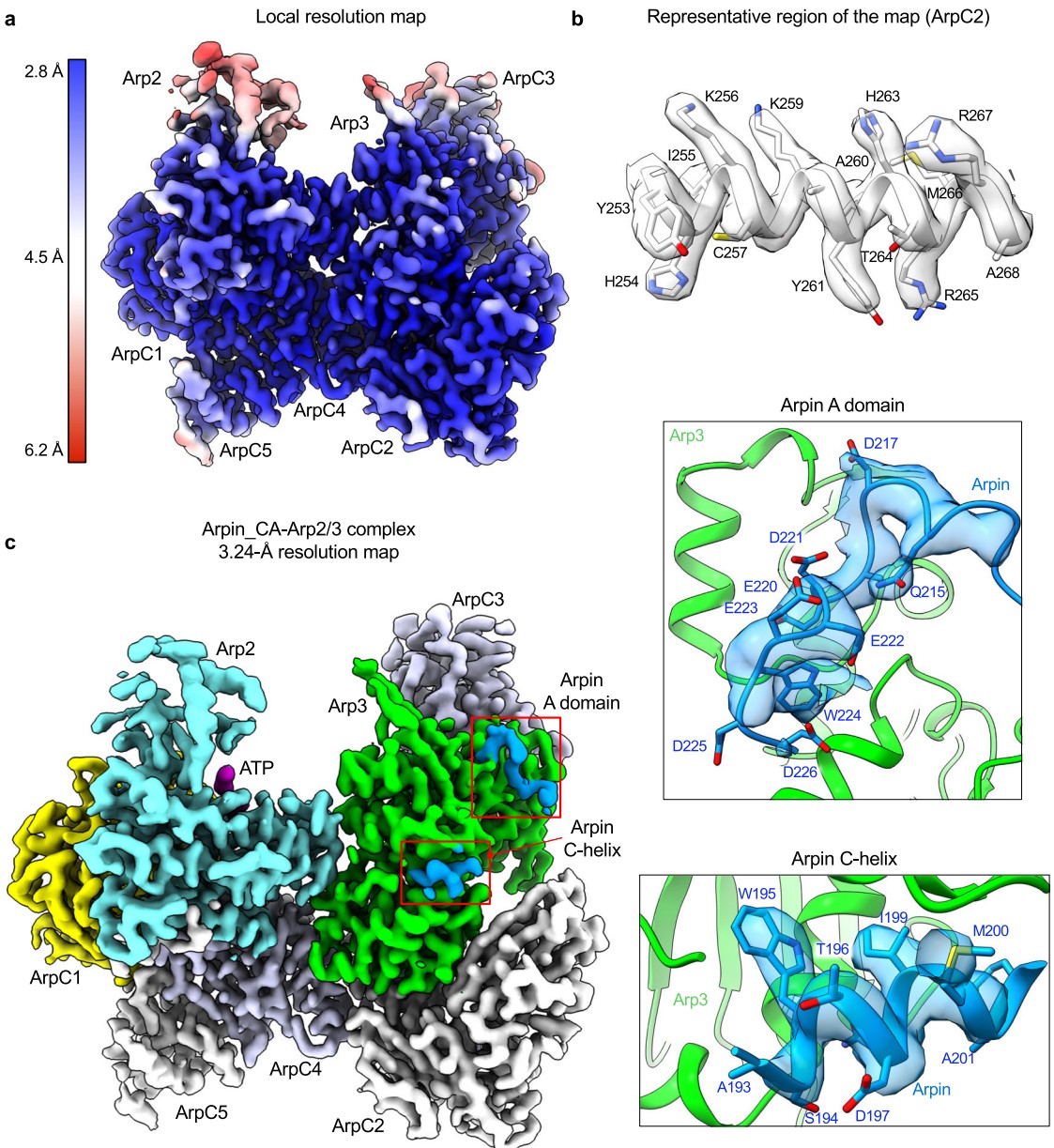

**Fig. 2 Cryo-EM structure of Arpin_CA bound to Arp2/3 complex. a** Cryo-EM map of Arpin_CA bound to Arp2/3 complex colored by resolution (as indicated by the side bar). Labels indicate the approximate location of Arp2/3 complex subunits. **b** Representative region of the 3.24-Å resolution cryo-EM map, showing the definition of side chains for the α-helix formed by residues Y253-A268 of ArpC2. **c** Cryo-EM map of Arpin_CA (marine blue) bound to Arp2/3 complex. Arp2/3 complex subunits are labeled and colored as follows: Arp2, cyan; Arp3, green; ArpC1, yellow; ArpC2, gray; ArpC3, blue-white; ArpC4, blue-white; ArpC5, gray. The Arpin_CA density appears as two separate patches shown in insets and corresponding to the C-helix and A domain.

When Arp3 is superimposed onto Arp2, a striking difference is observed in the positions of the C-helices of Arpin and N-WASP on these two subunits (Fig. 3a). Alternatively, we could attempt to superimpose the C-helix of Arpin onto that of N-WASP on Arp2. However, the C-helix of N-WASP binds differently to Arp2 and Arp3, occupying a different position and presenting a different face to each of the Arps[7]. Therefore, we do not know how to properly superimpose the C-helix of Arpin onto that of N-WASP. If the hydrophobic face of Arpin's C-helix is oriented toward Arp2, clashes are generated with the large side chain of Arpin's conserved residue W195. But, as we show below, W195 is not the sole determinant of Arpin's specificity for Arp3. On Arp3, in contrast, the CA region of Arpin superimposes remarkably well with that of N-WASP (Fig. 3b and Supplementary Fig. 3b), with

the contact surface on Arp3 being mostly limited to the two well-defined segments mentioned above (Fig. 3c). Arpin's A domain is nearly identical in sequence to that of N-WASP (Fig. 1a) and the two bind very similarly to Arp3 (Fig. 3d, f). Thus, the A domain is unlikely to be the source of the different activities of the two proteins. The linker between the C-helix and the A domain is 7-aa shorter in Arpin than in N-WASP (Fig. 1a). Yet, this linker is poorly defined in the structures of both N-WASP[7] and Arpin and does not directly contact Arp3 in the structure (Fig. 3c). Moreover, other NPFs such as WAVE also have a short linker. This leaves the C-helix as the most likely source of the differences between the inhibitor Arpin and the activator N-WASP. Like the C-helix of N-WASP, that of Arpin is an amphipathic helix (residues S194-G207) with its hydrophobic side facing the

**Table 1 Cryo-EM data collection, refinement, and validation statistics (Arpin_CA-Arp2/3 complex).**

| | |
|---|---|
| ***Data collection and processing*** | |
| Magnification | 105,000 |
| Voltage, keV | 300 |
| Total electron exposure, e−/Å² | 50 |
| Defocus range, µm | −0.75 to −2.00 |
| Pixel size, Å | 0.872 |
| Symmetry | C1 |
| Initial no. particles | 1,533,167 |
| Final no. particles | 182,324 |
| Map resolution, Å | |
|  FSC threshold 0.143 (0.5) | 3.24 (4.16) |
|  Resolution range, Å | 2.8–6.2 |
| ***Refinement*** | |
| Initial models used (PDB codes) | 1K8K, 6UHC |
| Resolution refined structure, Å | 3.35 |
|  FSC threshold | −0.5 |
| Map sharpening B-factor, Å² | −127 |
| Model composition | |
|  No. non-hydrogen atoms | 15,040 |
|  No. residues | 1885 |
|  No. ligands | ATP: 2, Mg: 2 |
| Correlation model vs. data | |
|  CC (mask, box, peaks, volume) | 0.77, 0.76, 0.76, 0.78 |
| R.m.s. deviations | |
|  Bond lengths, Å (# > 4σ) | 0.004 (0) |
|  Bond angles, ° (#> 4σ) | 0.631 (8) |
| Validation | |
|  MolProbity score | 1.86 |
|  Clashscore | 11.40 |
|  Rotamer outliers, % | 0.55 |
| Ramachandran plot | |
|  Favored, % | 95.87 |
|  Allowed, % | 4.13 |
|  Disallowed, % | 0.00 |
| ADP (B-factors) | |
|  Protein, Å² (min/max/mean) | 43.30/151.74/67.79 |
|  Ligands, Å² (min/max/mean) | 52.63/72.50/65.19 |
| Accession codes | |
|  EMDB | 22416 |
|  PDB | 7JPN |

hydrophobic cleft between subdomains 1 and 3 of Arp3 (Fig. 3e, g). A comparison between the C-helices of N-WASP and Arpin on Arp3 reveals that Arpin's C-helix is one turn shorter than that of N-WASP at the N-terminus (Fig. 3b). This difference is crucial because the longer helix of NPFs displaces the C-terminal tail of Arp3[7] (Fig. 3e, g, insets), which is known to play an inhibitory role by locking Arp2/3 complex in the inactive conformation[20]. Contrary to NPFs, Arpin stabilizes the position of the Arp3 tail in the inhibited state. Arpin residue W195 is primarily responsible for this stabilization. It inserts into a hydrophobic pocket formed by Arp3 residues V146, A150, W153, L163, M383, L384, V413, and F414 (Fig. 3e, inset). The latter two residues form part of the C-terminal tail of Arp3, and when mutated to aspartic acid the NPF-independent activity of budding yeast Arp2/3 complex increases [20]. Arpin residue W195 as well as Arp3 residues V413 and F414 are strictly conserved in mammals (Arpin is not present in yeast), suggesting that this is a conserved inhibitory mechanism (Supplementary Fig. 3c, d).

**Replacing Arpin's C-helix by that of N-WASP turns Arpin into an NPF.** Inspired by the structures and sequence analysis (Fig. 4a), we designed a series of Arpin mutants to understand the source of its inhibitory activity using the pyrene–actin

polymerization assay (Fig. 4 and Supplementary Fig. 4a, b). As previously reported[10], in control experiments full-length Arpin inhibited actin (2 µM, 6% pyrene-labeled) polymerization induced by Arp2/3 complex (5 nM) with N-WASP WCA (200 nM) in a concentration-dependent manner (Fig. 4b and Supplementary Fig. 4a). Human Arpin has a cysteine residue (C204) within the C-helix where most mammalian species have serine (Fig. 4a). C204 is exposed to the solvent in the structure and does not bind directly to Arp3 (Fig. 3e). Accordingly, mutating C204 to serine within full-length Arpin (Arpin_C204S) did not alter its inhibitory activity (Fig. 4b and Supplementary Fig. 4a). Because a cysteine residue at this position can cause unintended crosslinking, other constructs studied here contain this background mutation. Replacing W195 within the C-helix by aspartic acid (Arpin_W195D) abolished most of the inhibitory activity of full-length Arpin (Fig. 4b and Supplementary Fig. 4a), which is consistent with the important role of this amino acid in the structure (Fig. 3e). The residual inhibitory activity of this mutant can be attributed to the A domain, shown before to retain partial inhibitory capacity[10]. The structure also suggested that the globular domain of Arpin is not involved in Arp2/3 complex binding. Consistently, construct Arpin_CA, used in the high-resolution structure (Fig. 2), had approximately the same inhibitory activity as full-length Arpin over a range of concentrations (Fig. 4b and Supplementary Fig. 4a, b).

We then designed a series of hybrid constructs between N-WASP WCA and Arpin_CA (Fig. 4a and Supplementary Fig. 4c) to address the source of the different activities of these two proteins. The overall strategy was to introduce incremental changes in the Arpin sequence that would ultimately turn it into an NPF such as N-WASP. In control experiments, N-WASP WCA (200 nM) strongly activated actin (2 µM, 6% pyrene-labeled) polymerization induced by Arp2/3 complex (25 nM) (Fig. 4c). The first hybrid construct analyzed added the WH2 domain of N-WASP N-terminally to the CA region of Arpin (Hybrid_WH2). Hybrid_WH2 did not activate Arp2/3 complex (Fig. 4c). We previously reported that actin delivery to Arp3, which Hybrid_WH2 can presumably achieve, is essential for Arp2/3 complex activation[7], but seemingly not sufficient. Indeed, the inability of Hybrid_WH2 to activate Arp2/3 complex suggests that actin—NPF binding to the high-affinity site-1 on Arp2-ArpC1 and actin delivery to Arp2 are also essential for activation, but consistent with the structure (Fig. 2c) and ITC data (Fig. 1h) Hybrid_WH2 cannot bind to this site. Building upon Hybrid_WH2, the next construct had the short linker between the C-helix and A domain of Arpin substituted by the longer linker of N-WASP (Hybrid_WH2_Linker). This construct also failed to activate Arp2/3 complex (Fig. 4c). This was not a surprising result, since as stated above the linker is flexible and does not appear to contribute much to the interaction (Fig. 3c), and some NPFs also have shorter linkers than N-WASP (Fig. 4a). Building upon the latter construct, we then substituted residue W195 in the C-helix by isoleucine, the corresponding amino acid in N-WASP (Hybrid_WH2_Linker_W195I). This construct also failed to activate Arp2/3 complex (Fig. 4c), suggesting that other features of the C-helix of Arpin prevent binding to Arp2. We thus decided to replace the entire C-helix of Arpin by that of N-WASP, keeping only the linker and A domain from Arpin (Hybrid_WH2_C-helix). This construct activated Arp2/3 complex approximately to the same level as N-WASP WCA (Fig. 4c), strongly suggesting that the main source of the differences between Arpin and N-WASP is Arpin's unique C-helix that can bind Arp3 but not Arp2. This interpretation was conclusively validated using ITC; like N-WASP[16] but unlike other Arpin constructs (Fig. 1e–h) Hybrid_WH2_C-helix bound to two sites on Arp2/3 complex, a high- and a low-affinity site (Fig. 4d).

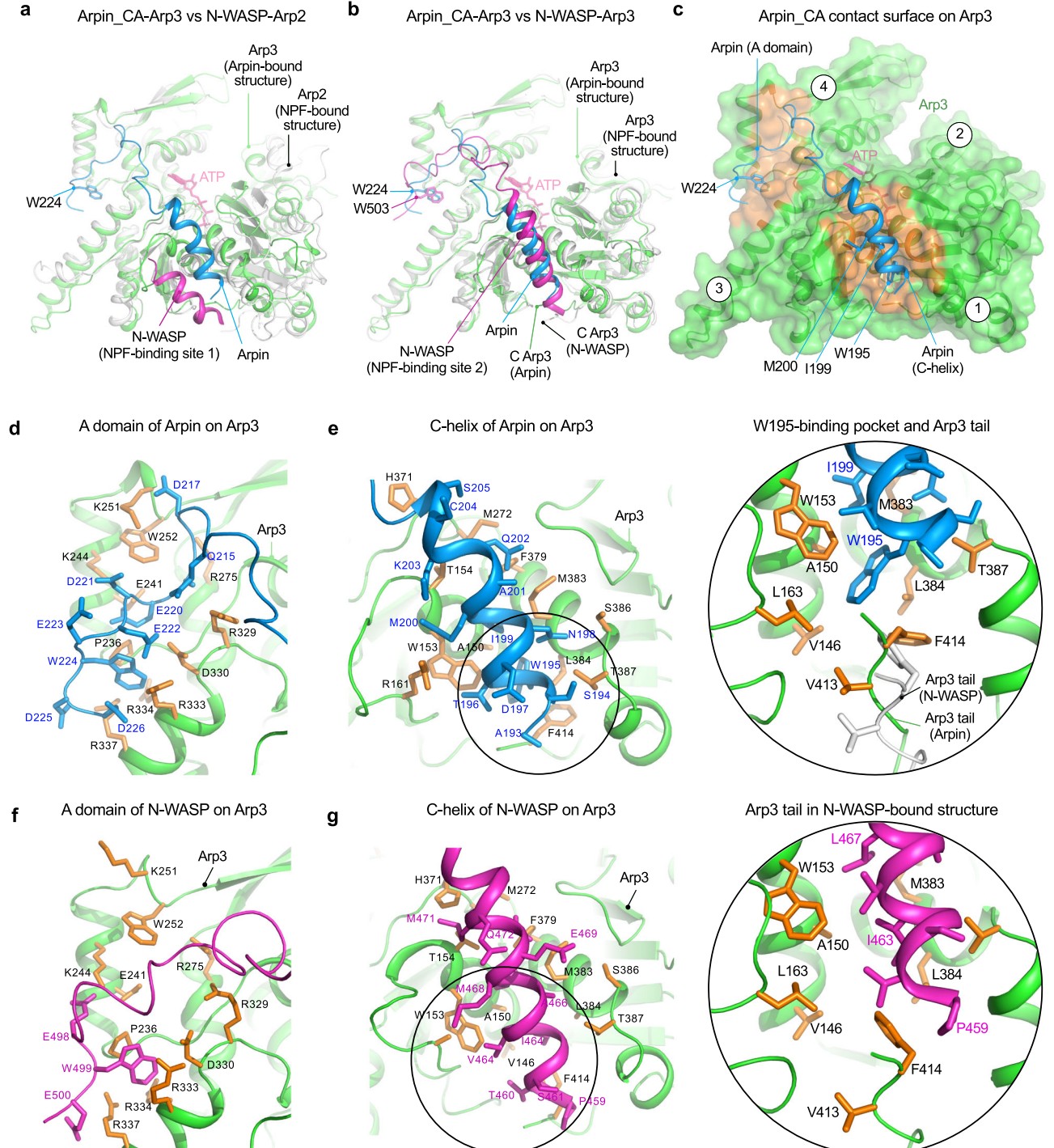

**Fig. 3 Interactions of Arpin and N-WASP with Arp2/3 complex. a**, **b** Superimposition of Arpin_CA-Arp3 (marine blue and green) onto N-WASP-Arp2 (**a**) and N-WASP-Arp3 (**b**) (magenta and gray). **c** The contact surface (orange) of Arpin on Arp3 involves Arpin's C-helix and the A domain, whereas the negatively charged linker between these two regions is flexible and does not participate in specific contacts. **d**–**g** Comparison of the specific interactions of the A domain (**d** and **f**) and C-helix (**e** and **g**) of Arpin and N-WASP with Arp3 (green with orange side chains). Insets show details of the interactions of the C-helices of Arpin (**e**) and N-WASP (**g**) with the C-terminal tail of Arp3. The inset in part e shows a comparison of the C-terminal tail of Arp3 in both structures (green-orange, Arpin structure; gray, N-WASP structure.

**Arpin C-helix mutants impair Arp2/3 complex binding and cell migration**. It was previously shown that Arpin's A domain was necessary for its inhibitory effect on lamellipodial dynamics in cells[10,21]. Since we showed above that Arpin also has a C-helix that is important for Arp2/3 complex binding and inhibition in vitro, we tested here the role of this helix in cells. Three FLAG-tagged Arpin constructs were transiently expressed in HEK-293T

cells: WT and mutants W195D and I199D/M200D that disrupt the hydrophobic face of the amphipathic C-helix. By Western blot analysis, Arp2/3 complex subunits co-immunoprecipitated with WT Arpin but were mostly absent with mutants W195D and I199D/M200D (Fig. 5a and Supplementary Fig. 5a), suggesting that the C-helix is essential for Arp2/3 complex binding in cells. We then tested the effect of these mutations in cell migration

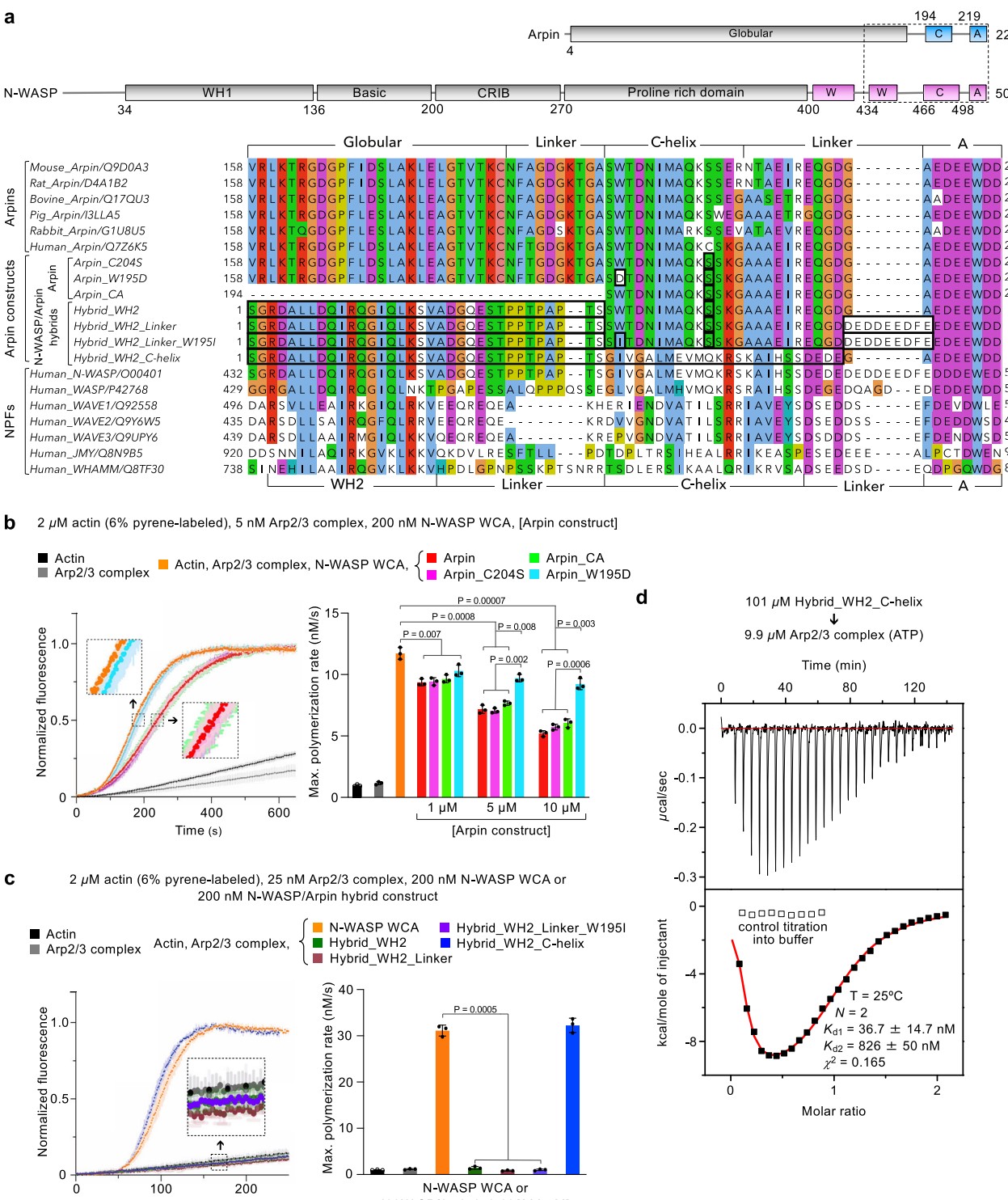

**b** 2 μM actin (6% pyrene-labeled), 5 nM Arp2/3 complex, 200 nM N-WASP WCA, [Arpin construct]

**c** 2 μM actin (6% pyrene-labeled), 25 nM Arp2/3 complex, 200 nM N-WASP WCA or 200 nM N-WASP/Arpin hybrid construct

**d** 101 μM Hybrid_WH2_C-helix → 9.9 μM Arp2/3 complex (ATP)

persistence. FLAG-tagged Arpin WT and mutants W195D and I199D/M200D were expressed in *ARPIN* −/− MCF10A cells (two different clones per expression condition were analyzed) and individual cells were tracked over time (Fig. 5b and Supplementary Fig. 5c–e). As previously reported[22], *ARPIN* −/− cells displayed increased migration persistence compared to parental wild-type cells. This phenotype is rescued by the expression of Flag-tagged WT Arpin but not mutants W195D and I199D/M200D. In agreement with the structural and biochemical data, these results confirm the importance of Arpin's C-helix for Arp2/

3 complex inhibition and the control of cell motility. The effect of Arpin knockout on other migration parameters, such as cell speed and mean squared displacement, depend on the cell type, and show little variation for MCF10A cells (Supplementary Fig. 5c–e).

## Discussion

The structurally most stable conformation of Arp2/3 complex is inactive, with the Arps splayed apart such that Arp2 partially blocks access of actin subunits to the barbed end of Arp3[14].

**Fig. 4 Replacing the C-helix of Arpin by that of N-WASP turns Arpin into an NPF. a** Domain diagrams of human Arpin and N-WASP (top). A dashed box highlights the C-terminal region included in the sequence alignment shown at the bottom. The alignment includes Arpin sequences from several species (upper portion) and several human NPFs (lower portion). UniProt accession codes are included with the name of each sequence. The middle portion of the alignment shows the human Arpin constructs and hybrid constructs of N-WASP and Arpin used in this study. Amino acids added or mutated relative to wild-type Arpin are boxed. **b** Left, time-course of actin polymerization by Arp2/3 complex activated by N-WASP WCA (protein concentrations are indicated), with and without Arpin constructs (10 μM). Data are shown as the average curve from three independent experiments with s.d. error bars in lighter color. Insets show a zoom of experimental curves that overlap. Right, maximum polymerization rates calculated from experiments analogous to those shown on the left (n = 3) and performed at three different concentrations of the Arpin constructs (as indicated, see also Supplementary Fig. 4a). **c** Left, time-course of actin polymerization by Arp2/3 complex activated by N-WASP WCA or hybrid N-WASP-Arpin constructs. The data are shown as described in part b, and the concentrations of the proteins are indicated. Right, maximum polymerization rates calculated from the experiments shown on the left (n = 3). For parts b and c, the statistical significance of the measurements was calculated using an unpaired, two-tailed t-test (P-values listed in the figure). The source data are provided as a Source Data file. **d** ITC titration of construct Hybrid_WH2_C-helix into Arp2/3 complex. The experimental conditions and fitting parameters are listed. Hybrid_WH2_C-helix is the only hybrid construct that activates Arp2/3 complex, and like N-WASP WCA binds to two sites on Arp2/3 complex.

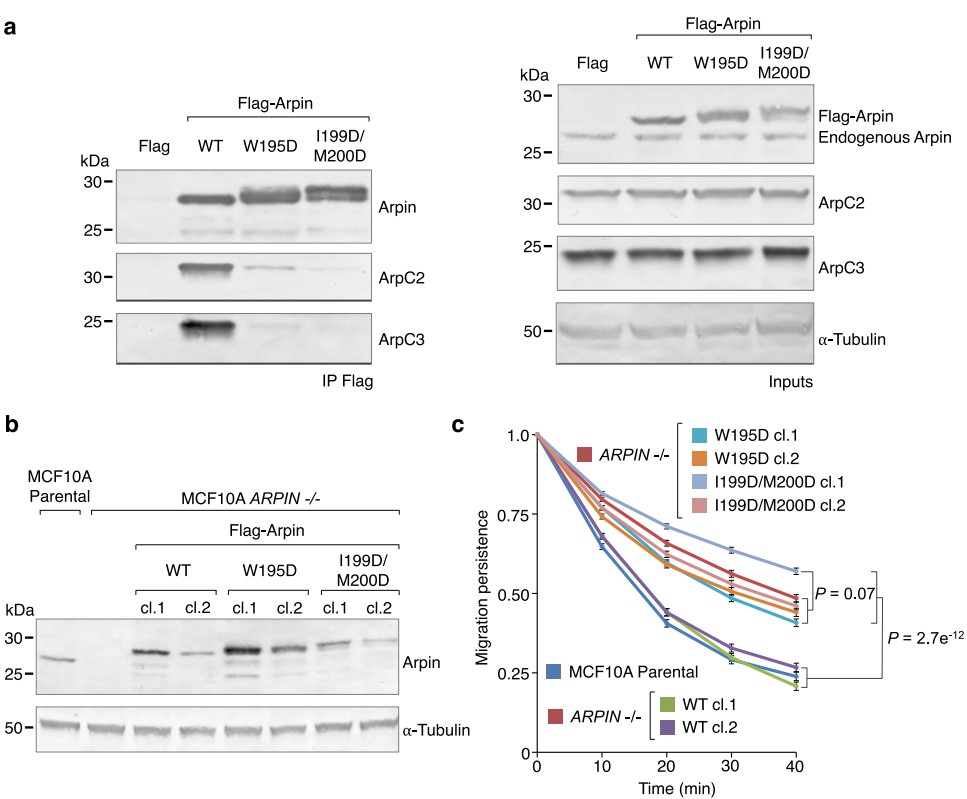

**Fig. 5 Arpin C-helix mutants impair Arp2/3 complex binding and cell migration. a** Immunoprecipitation of Arp2/3 complex subunits ArpC2 and ArpC3 from HEK-293T cells expressing Flag-tagged Arpin WT and mutants W195D and I199D/M200D. Right panel shows the loading controls of cell extracts and reveals both endogenous and exogenously expressed Arpin. **b** Western blot detection of Flag-tagged Arpin WT and mutants W195D and I199D/M200D expressed in *ARPIN* −/− stable MCF10A cell lines (loading control, α-tubulin). Arpin is not detected in control *ARPIN* −/− cells extracts. Uncropped and unprocessed blots are provided as a Source Data file. **c** Migration persistence of control parental and *ARPIN* −/− MCF10A cells compared to those of *ARPIN* −/− cells expressing constructs Arpin WT and mutants W195D and I199D/M200D (raw migration data in Supplementary Fig. 5c). The statistical significance of the measurements was calculated using a one-way ANOVA test (exact P-values are listed in the figure). Source Data are provided as a Source Data file. N = 74 cells for MCF10A parental, ARPIN −/− and WT cl.1, N = 72 cells for WT cl.2, N = 75 cells for W195D cl.1, N = 69 cells for W195D cl.2, N = 76 cells for I199D/M200D cl.1, and N = 67 cells for I199D/M200D cl.2. Results are expressed as means and standard error of the mean.

Biochemical work shows that interactions involving the C-terminal tail of Arp3, which is uniquely extended compared to that of actin and occupies most of the hydrophobic cleft at the barbed end of Arp3, help stabilize the inactive conformation of Arp2/3 complex[20]. To help understand how NPFs trigger activation, we recently determined the cryo-EM structure of N-WASP WCA bound to Arp2/3 complex[7]. The structure revealed that the CA region of N-WASP binds to two sites on Arp2/3 complex, one on Arp2-ArpC1 and one on Arp3, with the C-helix occupying the hydrophobic cleft at the barbed end of both Arps. An important revelation of the structure was that N-WASP binding to Arp3 shifts away the inhibitory tail of Arp3, which should in principle destabilize the inactive conformation. Yet, Arp2/3 complex in the N-WASP-bound structure remained in the inactive conformation, suggesting that releasing the tail of Arp3 is insufficient to trigger activation. Biochemical analysis further showed that activation requires actin delivery at the barbed end of both Arps, which likely explains why NPFs contain

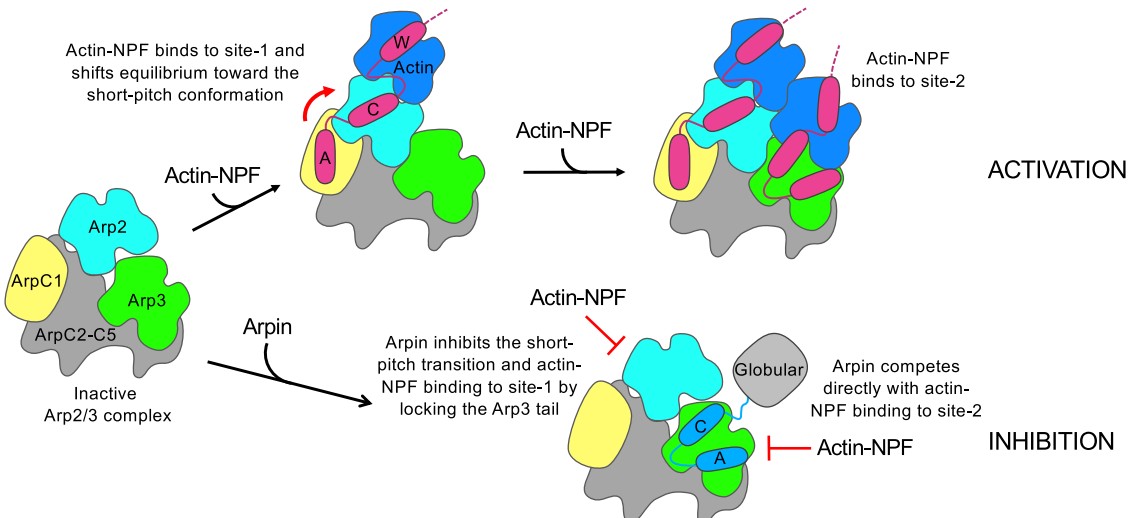

**Fig. 6 Model of Arp2/3 complex inhibition by Arpin versus activation by actin—NPF.** Arp2/3 complex is activated by actin—NPF binding to two sites, a high-affinity site-1 on Arp2-ArpC1 and a lower affinity site-2 on Arp3[7] (Activation, top). Actin-NPF binding to site-1 shifts the equilibrium towards the short-pitch conformation, which also facilitates actin—NPF binding to Arp3. Arpin inhibits Arp2/3 complex activation through a combination of two effects (Inhibition, bottom). By locking the inhibitory tail of Arp3[20], Arpin inhibits the short-pitch transition that actin—NPF binding to site-1 promotes. By binding to Arp3, Arpin directly competes with actin—NPF binding to site-2.

actin binding WH2 domains that precede the Arp2/3 complex binding CA region. It is within this context that we must analyze the mechanism of inhibition by Arpin uncovered here.

One finding, supported both by the structures and biochemical data (Fig. 1 and Supplementary Fig. 1), was that the N-terminal globular domain of Arpin does not directly participate in binding and inhibition of Arp2/3 complex. In the case of NPFs, domains N-terminal to the WCA region participate in regulation and subcellular localization[1]. We hypothesize the same may be true of Arpin's globular domain. Indeed, like the NPF WAVE, Arpin functions under Rac control, but the mechanism of regulation is unknown and is likely indirect[10]. Another unexpected finding was that like NPFs Arpin contains C-terminal C and A domains. Yet, unlike the CA region of NPFs that binds to two sites on Arp2/3 complex, Arpin's CA region binds only to the site on Arp3. Through a series of biochemical experiments, we were able to demonstrate that it is the C-helix of Arpin that restricts binding to Arp3 and not Arp2. A conserved tryptophan residue within the C-helix of Arpin (W195) is crucial for its inhibitory mechanism, by helping stabilize (or lock) the inhibited conformation of the C-terminal tail of Arp3 through hydrophobic interactions with conserved residues of the tail (V413 and F414). In other words, while NPFs release the Arp3 tail from its inactive conformation[7,20], Arpin has the opposite effect by stabilizing the tail in place. We thus propose a mechanism of Arpin-mediated inhibition of Arp2/3 complex that involves a combination of two effects (Fig. 6). First, by locking the inhibitory tail of Arp3 Arpin inhibits the short-pitch transition that actin—NPF binding to the first, high-affinity site-1 on Arp2-ArpC1 is known to promote[7]. Second, by binding to Arp3 Arpin competes directly with actin—NPF binding to the low-affinity site-2 on Arp3.

Our findings also reveal important differences and commonalities among NPFs and the inhibitor Arpin that could impact our understanding of other Arp2/3 complex regulators. Other A domain-containing regulators of Arp2/3 complex include the branch stabilizing protein cortactin[23], the Arp2/3 complex sequestering protein gadkin[24], and the proposed Arp2/3 complex inhibitor PICK1[25]. The structure of both Arpin and N-WASP bound to Arp2/3 complex suggests that the A domain can act as a "hook" to hold on to Arp2/3 complex in the branch in the case of

cortactin or to delocalize Arp2/3 complex in the case of gadkin. Unlike Arpin, however, PICK1 lacks a detectable C-helix adjacent to its A domain, and thus it is unlikely to function as an inhibitor. Indeed, we have found that PICK1 does not bind nor inhibit Arp2/3 complex in vitro[26]. Direct inhibition of Arp2/3 complex likely requires a mechanism to lock the Arps in the inactive conformation. Whether and how this is achieved by another proposed inhibitor, coronin[27], remains a mystery.

Finally, it is important to note that the hydrophobic cleft at the barbed end of actin constitutes a hot spot for interactions with numerous actin-binding proteins[28,29] and the clefts of the Arps are emerging as equally important in mediating interactions with Arp2/3 complex regulators. Established interactions of the hydrophobic clefts of the Arps now include the branch disassembly factor GMF that binds the cleft of Arp2[30], Arpin that binds the cleft of Arp3, and NPFs that bind the clefts of both Arps. The hydrophobic cleft in actin also mediates important contacts along the long-pitch helix of the actin filament[31,32]. A recent cryo-ET structure of Arp2/3 complex in the branch shows that the same is true of the clefts of the Arps[8]. Predictably, the structural characterization of other Arp2/3 complex regulators will reveal other interactions implicating the clefts of the Arps.

## Methods
**Proteins.** Arp2/3 complex was purified from bovine brain as previously described[17]. Briefly, frozen brains were homogenized in Arp buffer (20 mM HEPES [pH 7.5], 100 mM KCl, 1 mM MgCl₂, 1 mM EGTA, and 1 mM DTT) supplemented with protease inhibitors and clarified by centrifugation at 12,000 $g$ for 30 min. The supernatant was loaded onto a Macro-Prep High Q column (Bio-Rad) pre-equilibrated with Arp buffer. The flow-through, containing Arp2/3 complex, was applied onto a N-WASP WCA affinity column equilibrated with Arp buffer. Arp2/3 complex was eluted in 20 mM Tris [pH 8.0], 25 mM KCl, 400 mM MgCl₂, 1 mM EGTA and 1 mM DTT, concentrated and further purified through an SD200HL 26/600 column in Arp buffer. Actin was purified from rabbit skeletal muscle according to standard protocols. Briefly, actin was extracted from acto-myosin acetone powder with G-buffer (2 mM Tris (pH 8.0), 0.2 mM CaCl₂, 0.2 mM ATP, and 0.01% NaN₃), centrifuged at 20,000 $g$ for 30 min and polymerized with the addition of 50 mM NaCl and 2 mM MgCl₂. The F-actin pellet was homogenized in G-buffer with the addition of 10 mM DTT. After 1 h, actin was depolymerized by 3 day dialysis against G-buffer and centrifuged for 45 min at 277,000 $g$ to pellet any F-actin that did not depolymerize as well as any denatured actin. The pGEX plasmid encoding N-terminally tagged GST-TEV-Arpin (human) was described before[10], and used here as a template to obtain mutants C204S and

W195D using the Q5 mutagenesis kit (New England Biolabs) and to subclone construct Arpin_CA (residues 194–226). The gene encoding human N-WASP/human Arpin hybrid construct Hybrid_WH2 (Fig. 4a) was synthesized (Biomatik). Other N-WASP/Arpin hybrid constructs (Fig. 4a) were derived from this construct using the Q5 mutagenesis kit, except construct Hybrid_WH2_C-helix which was obtained using overlap-extension PCR (Supplemental Table 1 describes all the plasmids used in this study). These constructs were cloned between the NdeI and EcoRI sites of vector pTYB12 (New England BioLabs), comprising a chitin-binding domain for affinity purification and an intein domain for self-cleavage after purification. Sequences of primer oligonucleotides used in this study are provided in the Supplementary file.

All the protein constructs were expressed in BL21(DE3) cells (Invitrogen), grown in Terrific Broth medium at 37 °C until the $OD_{600}$ reached a value of 1.5–2. Expression was induced with 0.5 mM isopropylthio-β-galactoside (IPTG) and carried out for 16 h at 19 °C. Cells were harvested by centrifugation, resuspended in lysis buffer (50 mM Tris [pH 8.0], 50 mM NaCl, 1 mM EDTA, 2 mM DTT, 5% glycerol (v/v), and 100 μM phenylmethylsulfonyl fluoride) and lysed using a Microfluidizer apparatus (Microfluidics).

Cell lysates of Arpin wild-type and mutants C204S and W195D were incubated with glutathione-agarose resin (GoldBio) for 2 h at 4 °C (rocking). The lysis buffer was substituted with 50 mM Tris [pH 8.0], 100 mM NaCl, 2.5 mM $MgCl_2$, and 2 mM β-mercaptoethanol in three dialysis steps (0%, 50%, and 100%). On-column TEV cleavage was carried out overnight at 4 °C. TEV protease, containing an N-terminal His-tag was removed on a Ni-NTA agarose column (Qiagen). Proteins were additionally purified through a Superdex 200 26/60 size exclusion column (Pharmacia Biotech) and concentrated using a Vivaspin Turbo 3 kDa concentrator (Sartorius). Protein concentrations were measured using a UV/Vis spectrophotometer (GE Healthcare) ($\lambda = 200$–350 nm). The theoretical extinction coefficients at 280 nm for Arpin wild-type and mutant W195D are 29,450 and 23,950 $M^{-1}cm^{-1}$, respectively.

After affinity purification on a chitin column (New England BioLabs) and self-cleavage using dithiothreitol (2 days, 20 °C), all the hybrid constructs were purified by high-performance liquid chromatography (HPLC) using a reverse-phase C18 column and a 0–90% gradient of acetonitrile with 0.1% trifluoroacetic acid (TFA). Peak fractions were checked by MALDI-TOF mass spectrometry and subjected to three rounds of lyophilization/resuspension in 80% methanol to remove TFA. Proteins were stored as lyophilized powder at −20 °C and resuspended for use in 20 mM HEPES [pH 7.5] and 200 mM NaCl.

**Cryo-EM sample preparation and data collection**. Bovine Arp2/3 complex (8.6 μM) was mixed with human Arpin (2-fold molar excess) and the complex fraction used in cryo-EM grid preparation was isolated by glycerol gradient centrifugation (Supplementary Fig. 1a). Peak gradient fractions of the complex (as determined by Bradford and SDS-PAGE analyses) were concentrated to ~5 mg/ml. Samples were dialyzed against glycerol-free Arp buffer, supplemented with 0.2 mM ATP and 1 mM DTT. Samples were then diluted to 0.15 mg/mL with dialysis buffer and plunge frozen. Although we previously found that the use of detergent (NP-40) substantially improves the distribution of Arp2/3 complex particles[7], detergents were not used in this case because they caused the complex to be excluded from grid holes. Samples of the complex (2 μL) were applied to glow-discharged (1 min, easiGlow, Pelco) R2/2 300-mesh Quantifoil holey carbon grids (Electron Microscopy Sciences). The grids were blotted for 2 s with Whatman 41 filter paper and flash-frozen by plunging into liquid ethane using a Leica EM CPC manual plunger. EM grids were made in batches and the freezing conditions were optimized using an FEI TF20 microscope operating at 200 kV and equipped with an FEI Falcon III camera.

To obtain the structure of the complex with Arpin_CA, Arp2/3 complex (25 μM) was mixed with 5-fold molar excess Arpin_CA in Arp buffer. Unlike the sample with full-length Arpin, this sample tolerated well the use of detergents and cryo-EM grids were prepared with the addition of NP-40 to a final concentration of 0.0025%, which improves particle distribution[7]. Grids were made using 2 μL Arpin_CA-Arp2/3 complex, vitrified and screened as described above.

Cryo-EM datasets were collected automatically by Latitude (Gatan) on an FEI Titan Krios transmission electron microscope operating at 300 kV and equipped with a K3 (Gatan) direct electron detector with an energy quantum filter (Gatan). Images were taken at a nominal magnification of 105,000x in super-resolution mode, resulting in a pixel size of 0.436 Å. A total of 9661 images from a single Quantifoil R2/2 300-mesh grid (full-length Arpin sample) and 5599 images from a single Quantifoil R1.2/1.3 200-mesh grid (Arpin_CA sample) were collected at a defocus range of −0.75−−2.00 m. The exposure time was 2.6 sec, divided into 40 frames, at a dose rate of 14.2 $e^-$ $sec^{-1}$ $pixel^{-1}$, resulting in a nominal dose of 50 $e^-/Å^2$ (Table 1)

**Cryo-EM data processing**. Datasets were binned during motion-correction using MotionCorr2[33] resulting in a physical pixel size of 0.872 angstroms and imported into cryoSPARC (version v2.12.4)[34] for contrast transfer function (CTF) correction with CTFFIND4.1.13[35]. Cryo-EM datasets were processed as previously described[7], using a combination of cryoSPARC and Relion (version 3.0.8)[36].

For the full-length Arpin sample, micrographs were sorted based on CTF correction, and micrographs with a maximum resolution fit >8 Å were excluded, resulting in a subset of 8740 micrographs. CryoSPARC blob picking, followed by

2D classification were used to generate 2D classes for reference-based template picking. Template picking in cryoSPARC resulted in a total of 4,485,086 particles. Reference-free 2D classification in cryoSPARC was performed to remove particles that lacked secondary structure details (Supplementary Fig. 1b), resulting in a subset of 1,114,862 particles. Particles were then transferred to Relion for initial 3D model generation followed by 3D classification. Successive rounds of 3D classification yielded a final subset of 408,377 particles. Density corresponding to the globular domain of Arpin was not observed. To mitigate the effect of this missing density on the overall quality of the map, partial signal subtraction followed by 3D classification was performed in Relion using a mask covering the known regions of the structure[7]. This yielded a sub-class of 68,318 particles with improved density for the Arpin_CA region. 3D auto refinement of this class yielded a 3.6 Å resolution reconstruction (Supplementary Fig. 1c, d) that was post processed in Relion using a B-factor of −50.

Data processing of the Arpin_CA dataset was similar as to that of full-length Arpin. Motion corrected micrographs were imported into cryoSPARC (v3.2.0). Class averages from the full-length Arpin dataset were used as templates to pick 1,533,167 particles. 2D classification and subset selection yielded a total of 421,372 particles for subsequent processing. 2D classes showed a greater variety of orientations compared to the full-length Arpin sample, consistent with NP-40 reducing preferred orientation of vitrified Arp2/3 complex. An ab initio model was generated that was roughly consistent with the known structure of Arp2/3 complex at low resolution. Four rounds of heterogenous refinement, using the ab initio model and then the model resulting from each round as reference for the next round, yielded a map based on 195,415 particles. A final round of heterogenous refinement was performed to remove junk particles resulting in a final class of 182,324 particles. Iterative rounds of homogenous refinement and local CTF refinement in cryoSPARC yielded a final map at 3.24 Å resolution (half-map Fourier shell correlation, FSC = 0.143) displaying clear density for Arpin_CA. Post processing of the map was performed with deepEMhancer[37]. The orientation distribution of particles from the final reconstruction was determined using cryoEF[38] (Supplementary Fig. 2e) and the map was calculated to have an efficiency ($E_{od}$) of 0.78. The 3DFSC server (https://3dfsc.salk.edu/) was used to calculate the 3D-FSC of the final map, which shows high sphericity (0.974 out of 1) and a global resolution of 3.30 Å.

**Model building and refinement**. Model building and refinement were similar for the structures of full-length Arpin (3.6 Å resolution) and Arpin_CA (3.24 Å resolution) bound to Arp2/3 complex. The starting model for refinement was the original crystal structure of Arp2/3 complex determined at 2 Å resolution (PDB code: 1K8K)[14]. Half of Arp2 (subdomains 1 and 2) is missing in this structure and were modeled according to our recent cryo-EM structure of NPF-bound Arp2/3 complex[7], from where we also used the CA region of N-WASP bound to Arp3 as a stating model to build Arpin_CA. Arp2/3 complex subunits were individually fit to the map using rigid body refinement in the program Coot[39]. Multiple rounds of refinement using the program Phenix[40] and model building in Coot led to a final model with excellent stereochemical parameters, good map-to-model correlation, and a map-to-model FSC of 3.35 Å (Table 1 and Supplementary Fig. 2g). Maps for figures were low-pass filtered using the program e2proc3d within the EMAN2 suite[38] and figures were prepared using the programs Chimera[41], ChimeraX[42], and PyMOL (Schrödinger, LLC).

**Isothermal titration calorimetry**. ITC measurements were carried out on a VP-ITC apparatus (MicroCal). Full-length Arpin and Arp2/3 complex were dialyzed for 2 days against Arp buffer supplemented with either 0.2 mM ATP or ADP. The lyophilized Arpin_CA and Hybrid_WH2_C-helix peptides (Fig. 4a) were resuspended in the dialysis buffer prior to the experiments. Titrations were done at 25 °C and consisted of 27 injections (10 μL each) of the titrant in the syringe (as indicted in Figs. 1e–h and 4d), lasting for 20 s and with an interval of 300 s between injections. The concentration of the titrant (specified in the figures) was ~14-fold higher than that of the binding partner in the cell of 1.44 mL. The heat of binding was corrected for the exothermic heat of injection, determined by injecting ligand into the buffer. Data were analyzed using the program Origin (OriginLab Corporation).

**Actin polymerization assays**. Pyrene-actin polymerization assays were carried out using a Cary Eclipse fluorescence spectrophotometer (Varian) as previously described[7]. Before data acquisition, 2 μM Mg-ATP-actin (6% pyrene-labeled) was mixed with the indicated concentrations of Arp2/3 complex, N-WASP WCA, and/or Arpin and N-WASP/Arpin hybrid constructs in F-buffer (10 mM Tris [pH 7.5], 1 mM $MgCl_2$, 50 mM KCl, 1 mM EGTA, 0.1 mM $NaN_3$, and 0.2 mM ATP). Data were normalized to maximum polymerization rate using the equation: $S' = (S \times M_t)/(f_{max} - f_{min})$, where $S'$ is the apparent slope in μM/s, $S$ is the maximum slope of the raw trace, $M_t$ is the concentration of total polymerizable monomers, $f_{max}$ is the maximum fluorescence intensity at plateau, and $f_{min}$ is the baseline fluorescence intensity at the start of the reaction. The statistical significance of the measurements was determined using an unpaired two-way Student's $t$-test with the program Prism v7.0 (n.s, $0.05 < P$; *$0.01 < P < 0.05$; **$0.001 < P < 0.01$; ***$0.0001 < P < 0.001$; ****$P < 0.0001$).

**Plasmids and transfection for analysis in cells**. Arpin WT and mutants W195D and I199D/M200D were obtained using the QuikChange Lightning mutagenesis kit (Agilent). Full-length ORFs encoding Arpin WT and mutants W195D and I199D/M200D were cloned in our custom-made plasmid (MXS EF1Flag Blue2 SV40pA PGK Blasti bGHpA) between the FseI and AscI sites. For transient expression of Flag-Arpin WT, W195D and I199D/M200D, HEK-293T cells were transfected using Lipofectamine 3000 (Thermo Fisher Scientific). Stable MCF10A cells expressing these proteins were generated from MCF10A ARPIN −/− cells[22] transfected as described above and selected with 10 μg/ml Blasticidin (InvivoGen). Individual clones were picked with cloning rings. Sequences of primer oligonucleotides used in this study are provided in the Supplementary file.

**Antibodies**. Rabbit polyclonal antibodies obtained and purified using full-length Arpin were previously described[10]. Other antibodies were purchased, including ArpC2 pAb (Millipore #07-227), ArpC3 pAb (Sigma-Aldrich #ABN176), α-tubulin mAb (clone DM1A, Sigma #CP06).

**Cell cultures**. MCF10A cells were maintained in DMEM/F12 medium (Thermo Fisher Scientific) supplemented with 5% horse serum (Sigma), 100 ng/ml cholera toxin (Sigma), 20 ng/ml epidermal growth factor (Sigma), 0.01 mg/ml insulin (Sigma), 500 ng/ml hydrocortisone (Sigma) and 100 U/ml penicillin/streptomycin (Thermo Fisher Scientific). HEK-293T cells were maintained in DMEM medium (Thermo Fisher Scientific) supplemented with 10% fetal bovine serum (Thermo Fisher Scientific)

**Lysates and co-immunoprecipitation**. For expression analysis, transfected cells were lysed in 50 mM HEPES, pH 7.5, 150 mM NaCl, 1% NP-40, 0.5 % DOC, 0,1 % SDS, 1 mM CaCl$_2$ (RIPA). For co-immunoprecipitation, cells were lysed in 50 mM KCl, 10 mM HEPES pH 7.7, 1 mM MgCl$_2$, 1 mM EGTA, 1% Triton X100. Buffers were supplemented with EDTA-free protease inhibitor cocktail (Roche). Clarified lysates were incubated with 20 μl of anti-Flag coupled agarose beads (Sigma) for 3 h at 4 °C. Beads were washed in the same buffer and analyzed by SDS-PAGE.

**Western blots**. SDS-PAGE was performed using NuPAGE 4–12% Bis-Tris gels (Thermo Fisher Scientific). For Western blots, proteins were transferred using the iBlot system (Life Technologies) and developed using AP-coupled antibodies (Promega) and NBT/BCIP as substrates (Promega). Antibody dilutions were as follows: ArpC2 pAb (Millipore, Burlington, MA, USA) 1:1000, ArpC3 pAb (Sigma-Aldrich, St. Louis, MO, USA) 1:1000, α-Tubulin mAb (Sigma) 1:2000, Arpin 1:500. Uncropped and unprocessed blots are supplied in the Source Data file.

**Live cell imaging**. MCF10A cells were seeded onto glass bottomed μ-Slides (Ibidi) coated with 20 μg/ml of fibronectin (Sigma). Imaging was performed on an Axio Observer microscope (Zeiss), equipped with a Plan-Apochromat 20x/0.80 air objective, a Hamamatsu camera C10600 OrcaR2 and a PeCon XL multi S1 RED LS incubator (including heating unit XL S, Temp module, CO$_2$ module, heating insert PS, and CO$_2$ cover). Pictures were taken every 10 min for 24 h.

**Analysis of cell migration and statistical analysis**. Single-cell trajectories were tracked with Image J. The migration parameters, directional autocorrelation, mean square displacement, average cell speed, and single-cell trajectories plotted at origin were calculated using the DiPer suite[43]. Data from three independent experiments were pooled for analysis and plots. Results are expressed as the mean and standard errors of the mean (s.e.m.). The statistical analysis of migration persistence, measured as the movement autocorrelation over time, was performed using the program R and fit for each cell individually using an exponential decay with plateau, as described[21]. The fits were compared using one-way ANOVA analysis on nonlinear mixed-effect models for each condition (n.s, $0.05 < P$; *, $0.01 < P < 0.05$; **, $0.001 < P < 0.01$; *** $0.0001 < P < 0.001$; ****$P < 0.0001$).

**Reporting summary**. Further information on research design is available in the Nature Research Reporting Summary linked to this article.

## Data availability

The data that support this study are available from the corresponding authors upon reasonable request. Cryo-EM maps and models were deposited in the Electron Microscopy Data Bank EMDB-22416 and the atomic coordinates were deposited in the Protein Data Bank with accession code 7JPN. The models used in Fig. 1 are available in the PDB database under accession codes 6YW7 and 6UHC. The raw data for the pyrene-actin polymerization and cell migration experiments in this study are provided in the Source Data file. Source data are provided with this paper.

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

## Acknowledgements

Supported by National Institutes of Health grants R01 GM073791 and RM1 GM136511 to R.D., T32 GM008275 to T.V.E., and Agence Nationale de la Recherche grant ANR-20-CE13-0016-01 and Institut National du Cancer grant INCA_6521 to A.M.G. Data collection at the National Cancer Institute's National Cryo-EM Facility (NCEF) was supported by contract HSSN261200800001E. The use of computational recourses at the University of Pennsylvania was supported by NIH instrumentation grant S10OD023592.

## Author contributions

R.D. and A.M.G. conceived and directed the work. All the authors participated in the design and execution of the experiments and the analysis of the data. F.E.F. and R.D. wrote the paper and prepared the figures for publication.

## Competing interests

The authors declare no competing interests.
