## [Peer Review File · Nature Communications]

Reviewers' Comments:

Reviewer #1:

Remarks to the Author:

"Molecular mechanism of Arp2/3 complex inhibition by Arpin" Fregoso et al

In the work by Fregoso et al, the molecular mechanism for Arpin inhibition of the Arp2/3 complex is revealed using Cryo-EM reconstruction of the complex at 3.24 Å. Well-performed biochemical studies confirm and extend the findings in the structure. This work provides a different mechanism from previous reports and is of sufficient quality to supersede those previous studies. Specifically, the authors find that Arpin binds to and inhibits the Arp2/3 complex using a CA region. A globular domain, previously proposed to be involved was found to be dispensable for binding and inhibition. Next the authors turn to the question of how Arpin serves as an inhibitor instead of an activator. They make an interesting case that the C-helix of Arpin has evolved to bind only the Arp3 site, and it does so with a shorter C-helix. This is proposed to stabilize the inhibitory Arp3 C-terminal extension.

Overall, the work is convincing and warrants publication in Nature Communications. This has resolved the question of Arpin binding mechanism in my mind. While the audience for Arpin mechanism specifically is relatively small, this study also provides insight into the broader regulation of and allostery within the Arp2/3 complex.

I recommend acceptance with minor revisions.

I have several minor concerns.

Line 32) 'machinery' or 'machine'?

Line 140-143)

"If, alternatively, the C-helix of Arpin is superimposed onto that of N-WASP on Arp2, clashes are generated with Arp2, particularly involving the large side chain of Arpin's residue W195, which may in part explain why Arpin does not bind Arp2."

This resonated more than the structural alignment using the Arp3 positioning. This claim gets at the mechanism for why the Arpin CA is excluded from the Arp2/ArpC1 site. Can a figure panel be added to illustrate the claim directly?

Line 159-162) "Contrary to NPFs, Arpin stabilizes the position of the Arp3 tail in the inhibited state. Arpin residue W195 is primarily responsible for this stabilization. It inserts into a hydrophobic pocket formed by Arp3 residues V146, A150, W153, L163, L384, V413, and F414 (Fig. 3e, inset)."

I believe the claim that W195 stabilizes a hydrophobic cluster, and find that a compelling part of the overall story. However, the exact view used in the inset suggests that the Arp3 residues 415-418 (not resolved here, but present in reality) will collide with W195 setting up for a steric clash contradicting the claim. The depth cueing is quite subtle here and it took me a while to realize they are in different Z-positions in the inset view. Is there a better view to show the cluster?

Line 164-165) "Arpin residue W195 [is]... strictly conserved in Metazoa (Arpin is not present in yeast)"

Supp Figure 4d features only mammalian sequences. Claiming strict conservation in metazoans overstates the result from the alignment. Either obtain sequences from a wider evolutionary range, or adjust the claim to a more narrow evolutionary range.

Related issue, alignment and shading methods are not described.

Line 179-182) "Replacing W195 within the C-helix by aspartic acid (Arpin_W195D) abolished most of the inhibitory activity of full-length Arpin (Fig. 4b and Supplementary Fig. 4a), which is consistent with the important role of this amino acid in the structure (Fig. 3e)."

The use of hydrophobic to aspartate mutations throughout feels overly aggressive in some places. To say that loss of the C-helix is important, those are reasonable. But here, there is a claim about the tryptophan specifically. Does Arpin CA lose activity with the W195A mutation?

Reviewer #2:

Remarks to the Author:

This manuscript describes an excellent cryo-EM work determining the structure of the Arp2/3-Arpin complex. The authors find that Arpin binds only to the hydrophobic cleft of Arp3 through its CA domain. Careful side-by-side structural analysis of Arpin and N-WASP binding to the Arp2/3 complex provide convincing explanation of why the 2 proteins bind differently, and why their effect on Arp2/3 nucleation efficiency is opposite. These hypotheses are ultimately verified when replacing Arpin's C-helix by the one of N-WASP turns Arpin into a nucleation promoting factor.

I have no objection to the manuscript being published as is. I am only bothered that this study contradicts previously published results showing an interaction of Arpin with Arp2. Even though cryo-EM results have a much higher resolution, this is still surprising. As some authors have signed both studies, I think it would be necessary to discuss in greater detail the differences between the 2 studies, in order to understand what could have led to a different result before.

Minor comments :

- 1- Words such as "surprising" and "unexpected" are unnecessary.
- 2- The organization of the manuscript is sometimes a bit curious. Part of the introduction describing the Results could be shortened/removed. On the contrary, the beginning of the Result section would fit better in the Introduction. A slight reorganization would be beneficial to this manuscript.

Reviewer #3:

Remarks to the Author:

The regulation of the branching activity of the Arp2/3 complex is quintessential for cell locomotion. The manuscript from Dominguez group reveals the mechanism of inhibition of Arp2/3 complex by Arpin – a small protein involved in the control of directionality of cell migration. The paper provides a compelling data that fully support the authors conclusions and dramatically expands our understanding of the molecular mechanisms of Arp2/3 by NPFs. Therefore, I recommend the paper for publication in Nature Communications after the authors revise the paper as suggested below. Major suggestion: it is very hard to follow the last part of the discussion regarding the role/mechanism of action of multiple NPFs on the Arp2/3 and comparison of those with Arpin. Therefore, I strongly encourage the authors to add a cartoon that summarizes it.

Minor corrections:

1. P2: Missing "Introduction" header
2. P3 L87: "Indeed, this 86 density was very similar to that observed for NPF binding-site 2 on Arp3 in our recent cryo-EM 87 structure of N-WASP WCA (a fragment comprising the C-terminal WH2, C, and A domains of N88 WASP) bound to Arp2/3 complex 7." Should be corrected to " Indeed, this 86 density was very similar to that observed for NPF binding-site 2 on Arp3 in our recent cryo-EM 87

structure of N-WASP WCA (a fragment comprising the C-terminal WH2, C, and A domains of N88 WASP) bound to Arp2/3 complex 7."

3. P3 L88: Replace "Comparison of cryo-EM maps low-pass filtered at 5-Å resolution of Arp2/3 complex alone" with "Comparison of cryo-EM maps low-pass filtered to 5 Å resolution of Arp2/3 complex alone"

4. P3 L 93 "While the A domain of Arpin had been previously noted 10,12, the existence of the C domain was unexpected." The existence of C domain in the Arpin or the fact of its interaction with Arp3? Consider revision.

5. P4 L101 "verify the stoichiometry and binding affinity of the interaction" change to "verify the stoichiometry of Arpin binding and its affinity to Arp3"

6. P5 L134 change "filtered at" to "filtered to"

7. P36 In Fig 3 caption replace "b, The contact surface (orange)" to "c, The contact surface (orange)". Also, the extra repeat of Wasp helix should be outlined with an arrow in b. The displacement of the C-terminal tail of Arp3 by Wasp is not evident by comparison of panels e and g. Add superposition of the two and mark the difference with an arrow.

8. P28 Supl Fig 3 Replace "low pass filtered at " to "low pass filtered to"

9. P8 L255 Replace "Domains N-terminal to the WCA region of NPFs participate in regulation and subcellular localization, and the same could be expected of Arpin's globular domain." to "Since the N-terminal domains of NPFs have been shown to participate in their regulation and subcellular localization, we hypothesize that the globular N-terminal domain of Arpin has similar roles in the cell."

10. P8 L258 Replace " Another unexpected finding was that like NPFs Arpin contains not only an A domain but also a C domain. Yet, the CA region of Arpin binds only to one site on Arp2/3 complex, unlike NPFs that bind to two sites." with "Another unexpected finding was that despite the presence of both A and C domains in Arpin, it did not bind to both Arp2 and Arp3 subunits, but rather interacted specifically with the Arp3 only."

Response to reviewers' comments

General response

We would like to begin by thanking the three reviewers for their helpful comments and positive feedback. In the point-by-point response below, we use blue text for the reviewers' comments and black text for the authors' response. Nearly all the comments concerned minor changes in the text. In response to a comment from reviewer 3, we added a cartoon that summarizes the findings (Fig. 6) and a more detailed description of the inhibitory morel in the Discussion. For easy tracking, all the changes are highlighted yellow.

Reviewer 1:

I recommend acceptance with minor revisions

1. Line 32 'machinery' or 'machine'?

Replaced by "complex"

2. Line 140-143 "If, alternatively, the C-helix of Arpin is superimposed onto that of N-WASP on Arp2, clashes are generated with Arp2, particularly involving the large side chain of Arpin's residue W195, which may in part explain why Arpin does not bind Arp2." This resonated more than the structural alignment using the Arp3 positioning. This claim gets at the mechanism for why the Arpin CA is excluded from the Arp2/ArpC1 site. Can a figure panel be added to illustrate the claim directly?

This is an interesting question that shows our description of this point needed further clarification. As we described in Zimmet et al., 2020, the C-helix of N-WASP binds differently to Arp2 and Arp3. Not only is the position of the C-helix different for both Arps, but also the face this helix presents to the Arps is different. Therefore, we do not know how to fairly superimpose the Arpin C-helix onto that of N-WASP on Arp2. While Trp-195 is clearly a major difference, it is clearly not the only reason why Arpin does not bind to Arp2. Indeed, construct *Hybrid_WH2_Linker_W195I* (Fig. 4) in which Trp-195 was replaced by isoleucine (like in N-WASP) did not bind Arp2 and did not activate Arp2/3 complex. Binding to two sites (Arp2-ArpC1 and Arp3) and activation was only obtained when the entire C-helix of Arpin was replaced by that of N-WASP. Therefore, our description was incomplete, and we have changed the text accordingly (highlighted yellow). We thank the reviewer for this important comment.

3. Line 159-162 "*Contrary to NPFs, Arpin stabilizes the position of the Arp3 tail in the inhibited state. Arpin residue W195 is primarily responsible for this stabilization. It inserts into a hydrophobic pocket formed by Arp3 residues V146, A150, W153, L163, L384, V413, and F414 (Fig. 3e, inset).*" I believe the claim that W195 stabilizes a hydrophobic cluster and find that a compelling part of the overall story. However, the exact view used in the inset suggests that the Arp3 residues 415-418 (not resolved here, but present in reality) will collide with W195 setting up for a steric clash contradicting the claim. The depth cueing is quite subtle here and it took me a while to realize they are in different Z-positions in the inset view. Is there a better view to show the cluster?

We have slightly changed the orientation of the view in the inset to try to resolve this visual effect, but please note that the main view in Fig. 3e uses a different orientation that clearly shows that there is no collision between W195 and Arp3's C-terminus. It is always a challenge

to display atomic interactions and show everything to satisfaction, which is why we show two different orientations in the main figure and the inset. We also now added the two Arp3 tail conformations to Fig. 3e (inset).

4. Line 164-165) “*Arpin residue W195 [is]... strictly conserved in Metazoa (Arpin is not present in yeast)*” Supp Figure 4d features only mammalian sequences. Claiming strict conservation in metazoans overstates the result from the alignment. Either obtain sequences from a wider evolutionary range or adjust the claim to a more narrow evolutionary range. Related issue, alignment and shading methods are not described.

Changed to mammals and shading explained in figure legend.

5. Line 179-182 “Replacing W195 within the C-helix by aspartic acid (Arpin_W195D) abolished most of the inhibitory activity of full-length Arpin (Fig. 4b and Supplementary Fig. 4a), which is consistent with the important role of this amino acid in the structure (Fig. 3e).” The use of hydrophobic to aspartate mutations throughout feels overly aggressive in some places. To say that loss of the C-helix is important, those are reasonable. But here, there is a claim about the tryptophan specifically. Does Arpin CA lose activity with the W195A mutation?

The W195D mutation can be considered aggressive, but this was the goal. It was already known that Arpin contained an Acidic C-terminal domain and deletion of this domain indicated that it was essential for Arpin’s inhibitory activity (Dang et al., 2013). Here, we found that Arpin also contains a C-helix. To establish the importance of the newly found Arpin’s C-helix we devised mutations than should disrupt its ability to bind Arp3. A milder mutation would have had an intermediate phenotype, and thus would have been difficult to interpret. Note also that the W195D mutation was used by the Gautreau lab in cells, and we wanted the cell and biochemical data to talk to one another. Finally, as mentioned above (point 2), construct *Hybrid_WH2_Linker_W195I*, containing a milder W195I mutation, did not activate Arp2/3 complex and did not appear to bind Arp2. Therefore, the determinants of Arp2 vs. Arp3 binding are contained within the entire C-helix and not just W195 as explained in the manuscript.

Reviewer 2

This manuscript describes an excellent cryo-EM work determining the structure of the Arp2/3-Arpin complex ...

1. I have no objection to the manuscript being published as is. I am only bothered that this study contradicts previously published results showing an interaction of Arpin with Arp2. Even though cryo-EM results have a much higher resolution, this is still surprising. As some authors have signed both studies, I think it would be necessary to discuss in greater detail the differences between the 2 studies, in order to understand what could have led to a different result before.

We would like to thank the reviewer for the positive feedback. Concerning the previous study of Sokolova et al., 2017, as stated in our manuscript, these authors used low-resolution negative stain EM, known for its frequent artifacts and low resolution (~25Å). They also used human Arpin bound to yeast Arp2/3 complex, whereas yeast does not express Arpin. Furthermore, their 2D classes (Figure S5) show no extra density for Arpin, and show mostly one orientation (i.e. there is no 3D information as claimed). Clearly, the EM work of that paper is questionable. Cornering our own work, we used cryo-EM (not negative stain EM) to much higher resolution (3.24Å). We performed extensive biochemical analyses (ITC and pyrene-actin polymerization)

to validate the structural findings. While we clearly explain the different results of the two studies, as requested by the reviewer, we do not speculate about the reasons that may have led these authors to their incorrect results, which we feel is outside the topic of our paper. What is important in our opinion is that the scientific community will now learn the correct mechanism of Arpin-mediated inhibition.

2. Words such as “surprising” and “unexpected” are unnecessary.

While we try to limit the use of these words, they are sometimes useful to state that a result was neither anticipated nor previously suggested. We feel this is a matter that the editors of NC will surely address if there is a specific rule that prevents the usage of such words within a specific context.

3. The organization of the manuscript is sometimes a bit curious. Part of the introduction describing the Results could be shortened/removed. On the contrary, the beginning of the Result section would fit better in the Introduction. A slight reorganization would be beneficial to this manuscript.

Indeed, re-stating the main findings at the end of Introduction appears repetitive. Yet, this follows specific NC guidelines (please see *Guide to Authors*). Thanks again for the positive feedback.

Reviewer 3

I recommend the paper for publication in Nature Communications after the authors revise the paper as suggested below.

1. Major suggestion: it is very hard to follow the last part of the discussion regarding the role/mechanism of action of multiple NPFs on the Arp2/3 and comparison of those with Arpin. Therefore, I strongly encourage the authors to add a cartoon that summarizes it.

We thank the reviewer for this suggestion. We added a cartoon that summarizes the findings (new Fig. 6) and added text in the Discussion to support this cartoon.

Minor corrections:

2. P2: Missing “Introduction” header

Added “Introduction” header.

3. P3, L87: “Indeed, this density was very similar to that observed for NPF binding-site 2 on Arp3 in our recent cryo-EM structure of N-WASP WCA (a fragment comprising the C-terminal WH2, C, and A domains of N-WASP) bound to Arp2/3 complex.” Should be corrected to: “Indeed, this density was very similar to that observed for NPF binding-site 2 on Arp3 in our recent cryo-EM structure of N-WASP WCA (a fragment comprising the C-terminal WH2, C, and A domains of N-WASP) bound to Arp2/3 complex”

This could be a mistake since the original and newly proposed sentences are identical.

4. P3, L88: Replace “Comparison of cryo-EM maps low-pass filtered at 5-Å resolution of Arp2/3

complex alone” with “Comparison of cryo-EM maps low-pass filtered to 5 Å resolution of Arp2/3 complex alone”

“at” replaced by “to”, as suggested.

5. P3, L93 “While the A domain of Arpin had been previously noted, the existence of the C domain was unexpected” The existence of C domain in the Arpin or the fact of its interaction with Arp3? Consider revision.

Both were unexpected. We didn’t know Arpin had a C domain, and consequently we could not have predicted it bound Arp3 and less so that it would not bind Arp2. The sentence was reworded.

6. P4, L101 “verify the stoichiometry and binding affinity of the interaction” change to “verify the stoichiometry of Arpin binding and its affinity to Arp3”

Replaced by “verify the 1:1 stoichiometry of Arpin binding to Arp2/3 complex, establish the affinity of the interaction, and test the potential involvement of Arpin’s globular domain in binding to Arp2/3 complex”.

7. P5 L134 change “filtered at” to “filtered to”

“at” replaced by “to”, as suggested.

8. P36, in Fig. 3 caption replace “b, The contact surface (orange)” to “c, The contact surface (orange)”. Also, the extra repeat of N-WASP helix should be outlined with an arrow in b. The displacement of the C-terminal tail of Arp3 by N-WASP is not evident by comparison of panels e and g. Add superposition of the two and mark the difference with an arrow.

“b” replaced by “c”. We added the C-terminus of Arp3 (gray) to the inset in Fig. 3g to make the difference clear.

9. P28, Suppl. Fig 3 Replace “low pass filtered at” to “low pass filtered to”.

“at” replaced by “to”, as suggested.

10. P8, L255 Replace “Domains N-terminal to the WCA region of NPFs participate in regulation and subcellular localization, and the same could be expected of Arpin’s globular domain.” to “Since the N-terminal domains of NPFs have been shown to participate in their regulation and subcellular localization, we hypothesize that the globular N-terminal domain of Arpin has similar roles in the cell.”

Replaced by “In the case of NPFs, domains N-terminal to the WCA region participate in regulation and subcellular localization. We hypothesize the same may be true of Arpin’s globular domain”

11. P8, L258 Replace “Another unexpected finding was that like NPFs Arpin contains not only an A domain but also a C domain. Yet, the CA region of Arpin binds only to one site on Arp2/3 complex, unlike NPFs that bind to two sites.” with “Another unexpected finding was that despite the presence of both A and C domains in Arpin, it did not bind to both Arp2 and Arp3 subunits, but rather interacted specifically with the Arp3 only.”

Replaced by: “Another unexpected finding was that like NPFs Arpin contains C-terminal C and A domains. Yet, unlike the CA region of NPFs that binds to two sites on Arp2/3 complex, Arpin’s CA region binds only to the site on Arp3.”